# PRECISION AND RECALL REJECT CURVES FOR CLASSIFICATION

## ABSTRACT

For some classification scenarios, it is desirable to use only those classification instances that a trained model associates with a high certainty. To obtain such high-certainty instances, previous work has proposed accuracy-reject curves. Reject curves allow to evaluate and compare the performance of different certainty measures over a range of thresholds for accepting or rejecting classifications. However, the accuracy may not be the most suited evaluation metric for all applications, and instead precision or recall may be preferable. This is the case, for example, for data with imbalanced class distributions. We therefore propose reject curves that evaluate precision and recall, the recall-reject curve and the precision-reject curve. Using prototype-based classifiers from learning vector quantization, we first validate the proposed curves on artificial benchmark data against the accuracy reject curve as a baseline. We then show on imbalanced benchmarks and medical, real-world data that for these scenarios, the proposed precision- and recall-curves yield more accurate insights into classifier performance than accuracy reject curves.

## 1 INTRODUCTION

Today, machine learning (ML) models are used across a wide range of applications, where a common task is to train models for classification of objects. Many of these applications are safety-critical, such that the reliability and trustworthiness of classifications are particularly important. Examples of such applications come from the medical domain or high-stakes economic scenarios such as logistics for just-in-time production (Baryannis et al., 2019; Brintrup et al., 2019). A method to improve the reliability of classifiers is to estimate a certainty measure for each prediction and use only those predictions that were assigned a sufficiently high certainty. The combination of a certainty measure and a threshold that defines what degree of certainty is considered sufficient, has been termed *reject option* (Chow, 1970). Adding a reject option to a classifier allows to reject data points where the classification might be unreliable and can improve trust in the application (Artelt et al., 2022a; Sendhoff & Wersing, 2020; Wang et al., 2023).

Across classifiers, different certainty measures and thresholds may lead to different accuracies. Therefore, Nadeem et al. (2009) introduced accuracy reject curves (ARC) which display the accuracy as a function of the rejection rate for a given certainty measure and classifier. ARCs are a powerful tool for comparing different reject options and classifiers in many application scenarios. However, in some scenarios—instead of using the accuracy to judge classifier performance—precision and recall are preferable, most prominently, in imbalanced data sets. Thus the evaluation of a reject option using ARCs may be inappropriate. To close this gap, we here introduce reject curves for these alternative evaluation metrics, the precision reject curve (PRC) and the recall reject curve (RRC).

The structure of the paper is as follows. First, we review related work in Section 2. In Section 3 we briefly explain prototype-based classification, in particular classifiers from learning vector quantization, which are used within the experiments. Section 4 introduces the framework of reject options, followed by the introduction of existing and newly proposed PRC and RRC reject curves as evaluation techniques for reject options (Section 5). We demonstrate the usefulness of the PRC and RRC in experiments on artificial data with available ground-truth distribution, traditional benchmarks, and real-world medical data in Section 6. We close with the conclusion in Section 7.

## 2 RELATED WORK

Initially, reject options were introduced in the work of Chow (1970), who proposed a reject option with an optimal error-reject tradeoff if the class probabilities are known. Newer work introduced alternative names for classification with rejection, for example, *selective classification* (El-Yaniv & Wiener, 2010), *abstention* (Pazzani et al., 1994; Pietraszek, 2005), and *three way classification* (Yao, 2009). In order to evaluate classifiers with reject options, Nadeem et al. (2009) introduced ARCs, which are widely used today. Alternative approaches to evaluate classifiers with reject options were proposed and investigated in Hanczar (2019) and Condessa et al. (2017). For example, Hanczar (2019) propose to evaluate reject options by visualising different aspects, which allows additional perspectives when evaluating reject options, namely, finding a suitable trade-off between either error rate and rejection rate, cost and rejection rate, or true and false positives (receiver-operator trade-off). They claim that the latter one is less convenient as evaluation method compared to the other two.

Reject options can either be applied as a post-processing step in classification (Fischer et al., 2015; 2016) or can be integrated into the classifier itself, where the latter offers less flexibility compared to using reject options as post-processing, which can be applied to any classifier that allows to define a certainty measure. An example for integrated rejection is given in Villmann et al. (2016) and Bakhtiari & Villmann (2022) which builds a new type of classifier, the so-called classification by component networks (Saralajew et al., 2019).

Further, one can apply the same reject option across the whole input space, to obtain a so-called *global reject option*. On the other hand, a local reject option can be defined by setting one rejection threshold per class or for even more fine-grained partitions of the input space (Fischer et al., 2016). Global reject options with certainty measures suitable for prototype-based classification were introduced in Fischer et al. (2014a) and Fischer et al. (2015). To obtain certainty measures for prototype-based classifiers, probabilistic and deterministic approaches exist, where each approach shows advantages for certain data types (Fischer et al., 2014b). Local reject options and an efficient algorithm for determining optimal local thresholds are introduced in Fischer et al. (2016). The advantage of local reject options is that users can tune thresholds for individual classes or input space regions such as to increase the reliability for classes or regions of high relevance. Pillai et al. (2011) propose a generalisation for multi-label settings, which uses a $F1$-score instead of the accuracy as evaluation measure.

So far we addressed reject options for offline, batch-trainable classifiers. For online learning scenarios with drift, the authors in Göpfert et al. (2018) were the first to apply reject options. They show that a reject option with a fixed threshold does not increase the performance significantly and more sophisticated methods for choosing appropriate measures are needed.

Since it is not only important to reject unreliable decisions of a classifier, it may be of high importance why an input got rejected for classification. The authors of Artelt et al. (2022b) and Artelt et al. (2022a) propose first attempts to provide an explanation, where the latter work uses counterfactual explanations (Molnar, 2022). For a recent introduction to the topic of reject curves and related state of the art, see also Hendrickx et al. (2021). For a formal view on the topic, see Franc et al. (2023).

## 3 PROTOTYPE-BASED CLASSIFICATION

In this section, we will introduce prototype-based classifiers, which are used to demonstrate the usefulness of the proposed PRC and RRC in the following experiments. We consider prototype-based classifiers as this class of models has shown good performance on the considered example data and well-established certainty measures exist (Fischer et al., 2014a; 2015).

### 3.1 OVERVIEW OF PROTOTYPE-BASED CLASSIFIERS

We assume classification tasks in $\mathbb{R}^n$ with $Z$ classes, enumerated as $\{1, \ldots, Z\}$. Prototype-based classifiers are defined as set $W$ of prototypes $(w_j, c(w_j)) \in \mathbb{R}^n \times \{1, \ldots, Z\}$, and $j \in \{1, \ldots, J\}$ that are trained on example data $X$ to represent the data and its class borders.

Every prototype $w_j$ belongs to exactly one class with its class label $c(w_j) \in \{1, \ldots Z\}$. To classify a new data point $x$, the winner-takes-all-scheme is applied:

$$c(x) = c(w_l) \text{ with } w_l = \underset{w_j \in W}{\arg\min}\, d(w_j, x),$$

where $d$ is a distance measure, often the squared Euclidean distance. Any prototype-based model partitions the feature space into Voronoi cells with one responsible prototype per cell. A data point $x$ falling into a Voronoi cell is assigned the label of the related (closest) prototype, i.e. the winning prototype. The number of prototypes representing a class can be predefined for prototype-based models which leads to a sparse and interpretable representation of the given data $X$. Heuristics and cost function-based approaches are used as training techniques.

In the present work, we used extensions of the basic learning vector quantization algorithm (LVQ), proposed by Ritter & Kohonen (1989), which relies on a heuristic Hebbian learning paradigm. These extensions are the generalized matrix LVQ (GMLVQ) and local generalized matrix LVQ (LGMLVQ), and robust soft LVQ (RSLVQ), which we describe in detail in the following.

## 3.2 GMLVQ AND LGMLVQ

By formulating and optimizing explicit cost functions, extensions of LVQ are derived, namely, generalized LVQ (GLVQ) (Sato & Yamada, 1995) and RSLVQ (Seo & Obermayer, 2003) (described in the next section). For these models convergence guarantees can be given that follow directly from their derivation. For GMLVQ (Biehl et al., 2007), the distance metric is replaced by a general quadratic form, which is also learned during model training. The trained form represents a mapping that puts emphasis on the most discriminative input features and allows to reduce the feature set to the most relevant features only. The LGMLVQ (Schneider et al., 2009) adds a local metric to every prototype and has shown to outperform the GMLVQ in some scenarios.

Sato & Yamada (1995) proposed the GLVQ which was later extended to the GMLVQ and LGMLVQ. The GLVQ is based on the formalization as minimization of the cost function

$$E = \sum_i \Phi\left(\frac{d^+(x_i) - d^-(x_i)}{d^+(x_i) + d^-(x_i)}\right), \tag{1}$$

where $\Phi$ is a monotonically increasing function, e.g., the logistic function, and $d^+$ and $d^-$ are the distances to the closest prototype, $w^+$ and $w^-$, of the correct or incorrect class, for a data point $x_i$. GLVQ optimizes the location of prototypes by means of a stochastic gradient descent based on the cost function (Eq. 1). For a proof of the learning algorithm's validity at the boundaries of Voronoi cells see Hammer et al. (2005).

The GMLVQ generalizes the GLVQ to an algorithm with metric adaptation (Schneider et al., 2009). This generalization takes into account a positive semi-definite matrix $\Lambda$ in the general quadratic form which replaces the metric $d$ of the GLVQ, i.e. $d(w_j, x) = (x - w_j)^T \Lambda (x - w_j)$. The local version, the LGMLVQ, uses a single metric $d_j(w_j, x) = (x - w_j)^T \Lambda_j (x - w_j)$ for each prototype $w_j$.

## 3.3 RSLVQ

RSLVQ (Seo & Obermayer, 2003) assumes that data can be modeled via a Gaussian mixture model with labelled types. Based on this assumption, training is performed as an optimization of the data's log-likelihood,

$$E = \sum_i \log p(y_i | x_i, W) = \sum_i \log \frac{p(x_i, y_i | W)}{p(x_i | W)},$$

where $p(x_i | W) = \sum_j p(w_j) \cdot p(x_i | w_j)$ is a mixture of Gaussians with uniform prior probability $p(w_j)$ and Gaussian probability $p(x_i | w_j)$ centered in $w_j$ which is isotropic with fixed variance and equal for all prototypes or, more generally, a general (possibly adaptive) covariance matrix. The probability $p(x_i, y_i | W) = \sum_j \delta_{c(x_i)}^{c(w_j)} p(w_j) \cdot p(x_i | w_j)$ ($\delta_i^j$ is the Kronecker delta) describes the probability of a training sample under the current prototype distribution. For a given prediction $\hat{y}$, RSLVQ provides an explicit certainty value $p(\hat{y} | x, W)$, due to the used probability model at the price of a higher computational training complexity.

## 4   GLOBAL REJECT OPTION

A reject option for a classifier is defined by a certainty measure $r$ and a threshold $\theta$, which allows for individual samples to be rejected from classification if the classifier can not make a prediction with a certainty value above the threshold. The reject option is further called global if the threshold is constant across the whole input space, i.e., across all classes. (Extending the reject options proposed in the present work to local thresholds is conceivable but beyond the scope of this article (see Fischer et al. (2016) and Kummert et al. (2016)).)

Given a certainty measure

$$r : \mathbb{R}^n \to \mathbb{R}, x \longmapsto r(x) \in [0, 1]$$

for a data point $x$ and a threshold $\theta \in \mathbb{R}$, a reject option is defined as a rejection of $x$ from classification iff

$$r(x) < \theta.$$

A rejected data point will not be assigned with a predicted class label. All remaining, accepted data points with a certainty value higher or equal than $\theta$, we denote by $X_\theta$.

In our experiments we use the certainty measures Conf (2) and RelSim (3) that were proposed for prototype-based models in Fischer et al. (2014a). Additionally, for the artificial data we consider a Bayes classifier that provides ground-truth class probabilities and serves as a baseline (see below).

**Conf**   Classifiers based on probabilistic models such as RSLVQ provide a direct certainty value of the classification with the estimated probability $\hat{p}(\cdot)$.

$$r_{\text{Conf}}(x) = \max_{1 \le j \le Z} \hat{p}(j|x) \in (0, 1) \tag{2}$$

**RelSim**   The relative similarity (RelSim) (Fischer et al., 2014a) is based on the GLVQ cost function (1) and considers the distance of the closest prototype (the winner) $d^+$ and the distance of a closest prototype of any different class $d^-$ for a new unlabelled data point. The winner prototype with distance $d^+$ defines the class label of this new data point, if it is accepted. The measure calculates values according to:

$$r_{\text{RelSim}}(x) = \frac{d^- - d^+}{d^- + d^+} \in [0, 1]. \tag{3}$$

Values close to one indicate a certain classification and values near zero point to uncertain class labels. The values of $d^\pm$ are already calculated by the used algorithm such that no additional computational costs are caused. Furthermore RelSim (3) depends only on the stored prototypes $W$ and the new unlabelled data point $x$ and no additional storage is needed.

**Bayes**   The Bayes classifier provides class probabilities for each class provided the data distribution is known. The reject option corresponding to the certainty measure

$$r_{\text{Bayes}}(x) = \max_{1 \le j \le Z} p(j|x) \in (0, 1) \tag{4}$$

is optimal in the sense of an error-reject trade-off (Chow, 1970). We will use it as ground truth for an artificial data set with known underlying distribution. In general, the class probabilities are unknown, such that this optimum Bayes reject option can serve as Gold standard for artificially designed settings with a known ground truth, only.

## 5   EVALUATION OF REJECT OPTIONS USING REJECT CURVES

ARCs (Nadeem et al., 2009) are the state of the art for comparing classifiers with a reject option and show the accuracy of a classifier as function of either its acceptance or rejection rate. On the $x$-axis, ARCs show acceptance rates calculated as $|X_\theta|/|X|$, given an applied threshold $\theta$, while on the $y$-axis, the corresponding accuracy calculated on $X_\theta$ is shown. Similarly, the $x$-axis can show the rejection rate as $1 - |X_\theta|/|X|$.

ARCs can be easily calculated for binary and multi-class classification scenarios as long as an reject option can be defined for the classifier in question. Formally, the ARC for a given binary data set $X$ is defined as

$$ARC(\theta) : [0,1] \to [0,1], \frac{|X_\theta|}{|X|} \mapsto \frac{TP_\theta + TN_\theta}{|X_\theta|} \tag{5}$$

with $\theta \in \mathbb{R}^n$, and the true positives ($TP_\theta$) and the true negatives ($TN_\theta$) in $X_\theta$.

While for many classification tasks, in particular for balanced data sets, the accuracy and hence the ARC are suitable techniques, there are scenarios where other evaluation metrics of the classification performance are preferred. For instance, in highly imbalanced scenarios the accuracy of a classifier may be high simply due to—in the worst case—constantly predicting the majority class while the minority class is always misclassified. In such scenarios measures like the $F1$-score, or precision and recall (Van Rijsbergen, 1974) avoid misjudging the performance of a classifier on imbalanced data sets. In Pillai et al. (2011) a reject curve is proposed with the $F1$-score instead of the accuracy for multi-label settings. Similarly, we introduce the precision reject curve (PRC) and recall reject curve (RRC) as follows,

$$PRC(\theta) : [0,1] \to [0,1], \frac{|X_\theta|}{|X|} \mapsto \frac{TP_\theta}{TP_\theta + FP_\theta}, \tag{6}$$

$$RRC(\theta) : [0,1] \to [0,1], \frac{|X_\theta|}{|X|} \mapsto \frac{TP_\theta}{TP_\theta + FN_\theta}. \tag{7}$$

where $FP_\theta$ and $FN_\theta$ are the false positives and the false negatives in $X_\theta$. In this article we demonstrate the application of PRCs and RRCs for binary classification only. Analogously to ARCs for multi-class classification (e. g., Fischer et al. (2015)), both approaches can be extended to multi-class settings, as also precision and recall generalize to multi-class classification (Manning et al., 2009).

## 6 EXPERIMENTS

### 6.1 DATA SETS

To evaluate the proposed reject curves, we report experiments on an artificial data set, two common public benchmark data sets, and a real-world medical data set. For the artificial data set, class probabilities are known and we calculate the ground truth for reject curves using a Bayesian classifier. All data sets pose binary classification problems.

**Gaussian Clusters:**  The data set contains two artificially-generated, overlapping 2D Gaussian classes, overlaid with uniform noise. Samples are equally distributed over classes. Parameters used were means $\mu_x = (-4, 4.5)$ and $\mu_y = (4, 0.5)$, and standard deviations $\sigma_x = (5.2, 7.1)$ and $\sigma_y = (2.5, 2.1)$.

**Tecator data set:**  The goal for this data (Thodberg, 1995) set is to predict the fat content (*high* versus *low*) of a meat sample from its near-infrared absorbance spectrum. Samples are non-equally distributed over classes with $36.0\,\%$ versus $64.0\,\%$.

**Haberman's Survival Data Set:**  The data set contains $306$ instances from two classes indicating the survival of 5 years and more after breast cancer surgery (Dua & Graff, 2017). Data are represented by three attributes: *age*, the *year of operation*, and the *number of positive auxiliary nodes* detected. Samples were non-equally distributed ($26.5\,\%$ versus $73.5\,\%$).

**Adrenal:**  The adrenal tumours data set (Arlt et al., 2011) comprises $147$ samples composed of $32$ steroid marker values as features. The steroid marker values are measured from urine samples using gas chromatography/mass spectrometry. The data comprises two imbalanced classes, namely, patients with benign adrenocortical adenoma ($102$ or $68.4\,\%$ samples) and patients with malignant carcinoma ($45$ or $30.6\,\%$ samples). For medical details we refer to Arlt et al. (2011) and Biehl et al. (2012).

## 6.2 RESULTS

We demonstrate the usefulness of the PRC and the RRC for the three types of data sets, artificial Gaussian data, two benchmark data sets, and the Adrenal data set from a real-world medical application. We use a 10-fold repeated cross-validation with ten repeats for our experiments and evaluate models obtained by RSLVQ, GMLVQ, and LGMLVQ with one prototype per class. Since RSLVQ provides probability estimates, we use the certainty measure Conf (2) for rejection. In turn, GMLVQ and LGMLVQ lend itself to the certainty measure RelSim (3).

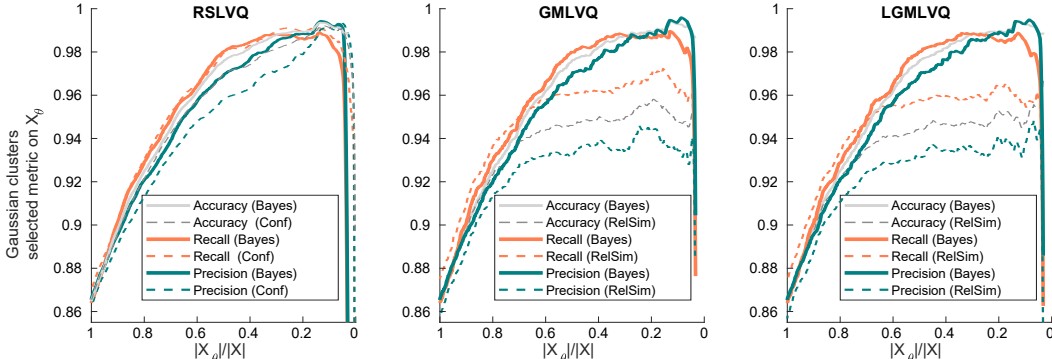

Figure 1: The image shows the averaged reject curves for the different LVQ models for the artificial Gaussian data. The solid lines represent the optimal classification performance of the Bayesian classifier. The PRCs and RRCs based on RelSim or Conf perform similar to the ARCs for the important regime of at least $80\%$ accepted data points. The ARCs are taken from Fischer et al. (2014a) and Fischer et al. (2016).

In Fig. 1, we show ARC, RRC, and PRC of the Bayesian classifier as well as for the trained prototype models for the Gaussian data set. The solid lines represent the optimal classification performance of the Bayesian classifier (mean over models in different runs). For the RSLVQ model, the PRCs and the RRCs resemble their respective baselines closely. Additionally, the prototype-based classifiers generate RRCs and PRCs that closely follow the baseline shape of the ARCs for nearly all acceptance rates, $|X_\theta|/|X|$. The latter is due to little noise and little overlap in the simulated data.

For the GMLVQ and the LGMLVQ the shapes of the ARCs, PRCs, and RRCs based on RelSim or Conf are similar to the respective Bayesian baseline results up to a rejection rate of $0.2 = (1 - |X_\theta|/|X|)$, i. e. an acceptance rate of $|X_\theta|/|X| = 0.8$. For lower acceptance rates ($|X_\theta|/|X| < 0.8$), all three reject options do not lead to substantial additional performance improvements. However, acceptance rates below $0.8$ may not be relevant for practical applications. In sum, our proposed reject curves mirror the optimal performance of the Bayesian classifier closely for acceptance rates that can be considered relevant for practical applications.

Fig. 2 shows reject curves for the benchmark data sets, where the ARCs of earlier work (Fischer et al., 2014a; 2016) are used for comparison of the RRCs and the PRCs. Precision is in the same range as accuracy in case of no rejection and for high and medium rejection rates ($|X_\theta|/|X| > 0.3$ for Tecator and $|X_\theta|/|X| > 0.1$ for Haberman). Recall has higher values in case of no rejection and behaves similarly to precision for rejection rates greater zero. Interestingly, for lower rejection thresholds $\theta$ the shapes of the RRCs and the PRCs are monotonically decreasing. This effect is most prominent for the Tecator data set. Such a behavior can be expected since precision and recall focus on one of the two classes instead of both (in case of binary settings). Here we see that accuracy is unable to evaluate the model performance meaningfully. Instead, PRC and RRC allow to evaluate model performance for a specific threshold $\theta$ with respect to the more appropriate measures recall and precision.

Reject curves are particularly relevant for safety-critical scenarios that are, for example, often encountered in the medical domain. Therefore, in our last experiment we demonstrate the application of the proposed reject curves on a real-world medical data set (Fig. 3). Here, we observe similar results as for the benchmark data sets—all reject curves reveal that from a certain value for $\theta$, precision and recall decline while accuracy keeps improving and thus offers an overly optimistic assessment of

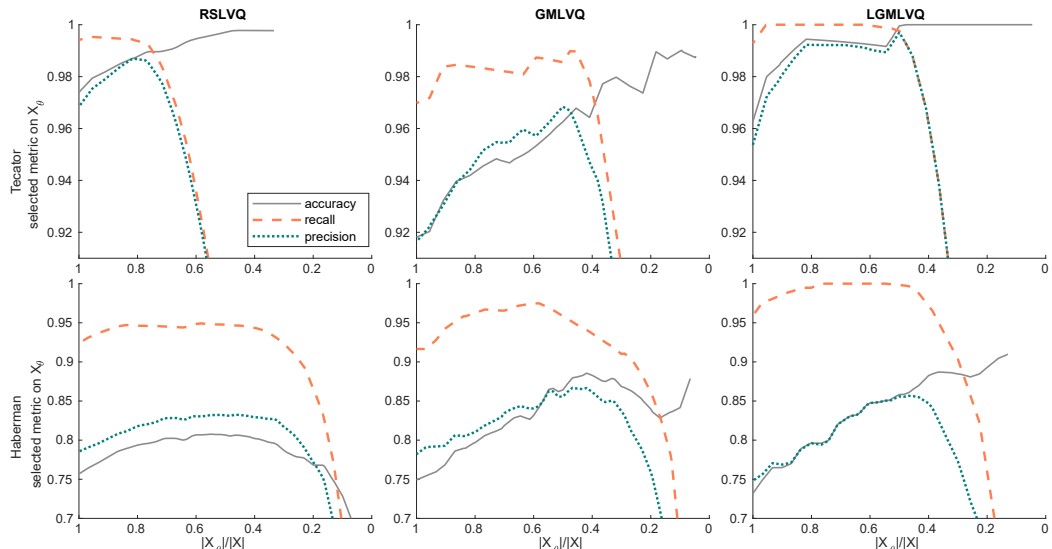

Figure 2: The image shows the averaged curves for the different LVQ models for the benchmark data sets. The ARCs are taken from Fischer et al. (2014a) and Fischer et al. (2016) and serve as a comparison. The PRCs and the RRCs based on RelSim or Conf perform differently for the given set-ups. This reveals interesting insights for the user in order to chose a suited reject threshold for the application scenario at hand.

low acceptance rates. We conclude that PRC and RRC enable users to select the most suited rejection threshold for applications with imbalanced data.

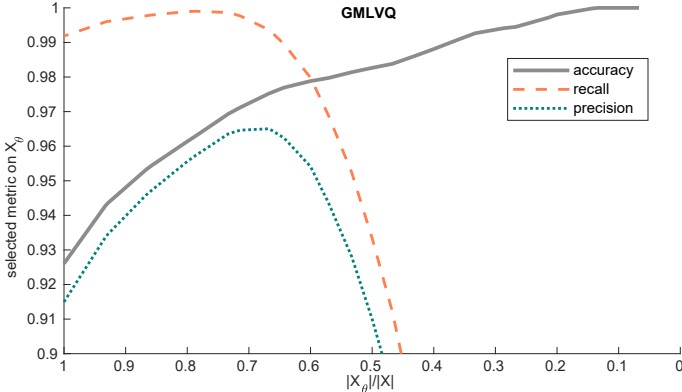

Figure 3: The image shows the averaged curves based on RelSim for the GMLVQ models for the adrenal data sets. The curves of the ARC and PRC perform similar in the important regime of at least 80% accepted data points while the RRC has a different shape. The ARC is taken from Fischer et al. (2016).

# 7 CONCLUSION

In this paper we introduced the precision reject curve (PRC) and the recall reject curve (RRC) to introduce techniques to evaluate reject options for classification tasks where precision and recall are the preferred evaluation metrics over the accuracy (e. g., for imbalanced data sets). We compare our proposed approach against the state-of-the-art evaluation using accuracy reject curves (ARC). To demonstrate the suitability of the proposed PRC and RRC, we applied both methods, first, to an artificial data set where we obtained a performance close to ground-truth solutions obtained from

Bayesian classifiers. Further, we applied our approach to two popular classification benchmarks, where we showed that our proposed approach allows additional insights into the performance of a classifier on imbalanced data, which could not be obtained from classical ARCs. Last, we applied the PRC and RRC to one real-world data set from the medical domain where trust in the classification result is particularly important. The latter experiment demonstrates the applicability of our approach as well as its usefulness in a real-world application domain with imbalanced data.

In sum, our results show that the ARC may be misleading for imbalanced data sets. Instead the PRC and the RRC provide trustworthy comparisons of reject options for classification results on imbalanced data. Future work may extend the proposed approach to multi-class classification and other evaluation metrics, e. g. true positive and false positive rates.

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
