# OpenReview forum: "Precision and Recall Reject Curves for Classification"
_ICLR.cc/2024/Conference — Submitted to ICLR 2024_

### Official Review · Reviewer_EkHi · 2023-10-15

**Soundness:** 3 good
**Presentation:** 2 fair
**Contribution:** 1 poor
**Rating:** 1
**Confidence:** 4

**Summary:**

This submission focuses on the classifcation scenario with rejection option where accuracy-reject curve is usually used to evaluate and compare the performance of different certainty measures over a range of thresholds for accepting or rejecting classifcations.

**Strengths:**

1. The motivation of this paper is reasonable, i.e., using precision-reject curve and recall-reject curve in the class-imbalance problems.

2. Some experiments are performed to show the differences of the precision-reject curve and recall-reject curve from accuracy-reject curve.

3. Some discussions are given.

**Weaknesses:**

1. The contribution is very limited. Precision-reject curve and recall-reject curve are two intuitive extensions of accuracy-reject curve. It is not suggested to presented them independently in one paper. It will be better to present them as part of one paper that proposes a classifcation algorithm with rejection option.

2. It is unclear why precision and recall are used. For class-imbalanced problems, F1 is the more commonly used measure.

3. In experiments, authors should pay more attentions to the difference of precision and recall from accuracy (e.g., figure 2). For figure 1, it is hard to observe their differences.

**Questions:**

If authors disagree my comments in weaknesses, clarifications can be presented in the rebuttal phase.

---

### Official Review · Reviewer_YWY1 · 2023-10-27

**Soundness:** 1 poor
**Presentation:** 3 good
**Contribution:** 1 poor
**Rating:** 3
**Confidence:** 4

**Summary:**

The article considers the problem of performance measurement under model uncertanty and extends a previously proposed method -- accuracy-rejet curves -- to precision and recall metrics rather than accuracy.

**Strengths:**

Rejection options is an important area of research, particularly impactful in some domains like medicine.

**Weaknesses:**

It is unclear how this is different from standard precision recall curves. In the binary classification case, given that precision and recall focus on the positive class only this work appears to simplify down to precision-recall curves. It therefore does not appear to propose anything novel that would help with choosing particularly thresholds for a task. It may be different in the case of multiple classes but none of  the experimental datasets have more than two classes.

**Questions:**

How is this different from standard precision/recall curves? It seems equivalent since precision/recall focus on the positive class only and disregard the negative class.

---

### Official Review · Reviewer_uffc · 2023-10-31

**Soundness:** 1 poor
**Presentation:** 3 good
**Contribution:** 1 poor
**Rating:** 1
**Confidence:** 4

**Summary:**

Accucracy-Reject-Curves (ARCs) are frequently used for evaluating classifiers with a reject option, i.e., where the classifier can choose to not classify an example. The authors argue that for imbalanced datasets, precision and recall reject curves, and evaluate them on four small datasets.

**Strengths:**

The assumptions upon which the work rests are valid, and it is quite likely that precision and/or recall reject curves are a good alternative to ARCs on imbalanced datasets.

**Weaknesses:**

Having said this, this extensions is a very trivial extension, as it is well-known (as the authors also observe) that precision and recall (and F1) are better evaluation measures for imbalanced data. So this paper could, in my opinion, only be of interest if the performed evaluation is exceptionally thorough and valid, so that this is a paper that clearly establishes the value of this proposal. Unfortunately, I think the evaluation is lacking for several reasons:
- the comparison is only on 4 small datasets, one of them as simple as two Gaussian clusters. The datasets are also rather small (less than 1000 instances), and - contrary to the claims of the paper - *not* heavily imbalanced (no class has less than 25% examples).
- no systematic evaluation of the data space, such as comparisons for several different rates of unbalancedness
- it is not clear why precision and recall (which are separate curves) have been selected, and not F1-reject curves, where one would only have to deal with one curve. These are, as the authors mention, also not new (although they have been proposed in a somewhat different context)
- there are some obvious benchmark methods that could/should have been included. For example, a natural extension would be to use local reject options, which have different thresholds for the majority and minority class. What is the advantage of precision-reject curves over ARCs in such settings? In the limiting case, when the minority class has a regular threshold, and the majority class has a fixed threshold of 1 (i.e., is never predicted), wouldn't you get something like a precision-reject curve for the minority class as well?

Minor:

The paper seems to be adapted from a version that used numeric citations. With author-year citations, constructs such as "The authors of Artelt et al. (2022) show.." are not good, this can be simply said as "Artelt et al. (2022) show...".

**Questions:**

None.

---

### Official Review · Reviewer_zM2w · 2023-10-31

**Soundness:** 2 fair
**Presentation:** 3 good
**Contribution:** 1 poor
**Rating:** 1
**Confidence:** 4

**Summary:**

In many real-world classification problems, one may want to provide a classifier the option to reject (or abstain from making a prediction) on samples that it is not certain about. In such cases, it is common to evaluate the trade-off between accuracy and rejection costs by plotting the accuracy as a function of rejection rates. However, in applications where there is severe class imbalance, plotting accuracy vs rejection rate may not be the best approach to evaluate the classifier. The paper proposes to instead use plots of precision and recall as a function of rejection rate, and presents some experiments to argue why these are better alternatives.

**Strengths:**

- The paper is well-written and does a good job of covering prior work on classification with rejection.
- Exploring alternate ways for evaluating a classifier with a reject option is an important problem to tackle.

**Weaknesses:**

- The paper is limited in novelty. The fact that metrics based on precision and recall are better alternatives to accuracy under class imbalance is well known in the literature. Merely proposing their use in evaluating rejection-based classifiers does not make for a significant or novel contribution.
- The experimental conclusions aren't very strong either. It is interesting that precision and recall curves show different trends compared to accuracy curves, but this observation alone does not make for a strong contribution. Besides, the datasets used are all small scale, containing a few hundred samples.

**Questions:**

- In Figures 2-3, as you reject more samples, it looks like both precision and recall show a downward trend. I had initially expected one of these metrics to be favored more compared to the other. Is the downward trend because we reject more from the minority class compared to the majority class?
- At 100% rejection rate (i.e. 0% acceptance rate), shouldn't all the metrics reach 100%. Wouldn't that be the more natural default value when all samples are rejected?

---

### Meta-Review · Area_Chair_j2FD · 2023-12-05

**Metareview:**

All reviewers agree that this submission cannot be accepted for the conference. The authors did not submit any rebuttal somehow confirming the evaluation of the reviewers.

**Justification For Why Not Higher Score:**

All reviewers agree that this submission cannot be accepted for the conference.

**Justification For Why Not Lower Score:**

N/A

---

### Decision · Program_Chairs · 2024-01-16

Reject